# Cyrene: A Green Solvent for the Synthesis of Bioactive Molecules and Functional Biomaterials

**DOI:** 10.3390/ijms232415960

**Published:** 2022-12-15

**Authors:** Andrea Citarella, Arianna Amenta, Daniele Passarella, Nicola Micale

**Affiliations:** 1Department of Chemistry, University of Milan, Via Golgi 19, I-20133 Milano, Italy; 2Department of Chemical, Biological, Pharmaceutical and Environmental Sciences, University of Messina, Viale Ferdinando Stagno D’Alcontres 31, I-98166 Messina, Italy

**Keywords:** cyrene, green chemistry, green synthesis, sustainable chemistry, nanomaterials

## Abstract

In the panorama of sustainable chemistry, the use of green solvents is increasingly emerging for the optimization of more eco-friendly processes which look to a future of biocompatibility and recycling. The green solvent Cyrene, obtained from biomass via a two-step synthesis, is increasingly being introduced as the solvent of choice for the development of green synthetic transformations and for the production of biomaterials, thanks to its interesting biocompatibility, non-toxic and non-mutagenic properties. Our review offers an overview of the most important organic reactions that have been investigated to date in Cyrene as a *medium*, in particular focusing on those that could potentially lead to the formation of relevant chemical bonds in bioactive molecules. On the other hand, a description of the employment of Cyrene in the production of biomaterials has also been taken into consideration, providing a point-by-point overview of the use of Cyrene to date in the aforementioned fields.

## 1. Introduction

The synthesis of biologically active molecules remains one of the most discussed topics among organic chemists, and the development of green approaches for the construction of new chemical bonds has become an interesting growing task. At the same time, the interplay between biomaterials and renewable resources has provided a window of opportunity for the development of novel eco-friendly nanomaterials. Industrial and laboratory processes are characterized by the accumulation of chemical waste, which are mainly represented by residues of reaction solvents. Today, green chemistry is deeply engaged in the search for eco-friendly and sustainable approaches in organic synthesis and for the production of functional materials, mainly involving the use of green solvents, with a high safety profile and low environmental impact. However, the search for alternatives to conventional organic solvents remains an arduous and difficult task. In 2014, the green solvent Cyrene, obtained from biomass and produced by Circa, was used for the first time as a solvent for organic synthesis reactions [1]. From then on, this solvent has come into use as a replacement for DMF (*N,N*-dimethyl formamide), DMSO (dimethyl sulfoxide), NMP (*N*-methylpyrrolidone) and DMAc (*N,N*-dimethyl acetamide), and a considerable number of articles elected it as a valid green alternative, considering it an environmentally friendly choice compared to the aforementioned toxic solvents. In particular, it appeared to be a valid substitute for DMF in carrying out amide bond synthesis reactions, known to be one of the most important and challenging transformations in organic chemistry because of the widespread occurrence of amides in modern pharmaceutical and biologically active compounds. In this context, Cyrene appeared to be the green solvent of choice for the synthesis of amides and peptides. Furthermore, it has recently also been employed in the synthesis of polyesters and in the development of nanoparticles, and hence its use is expected to grow in the future. Cyrene has also been used as a vehicle solvent in antimicrobial susceptibility testing, resulting to be a noteworthy substitute to the common DMSO [2], and as a solvent for the extraction of pharmaceutically important curcuminoids from turmeric with ultrasound-assisted procedures, demonstrating excellent performance and product recovery [3]. Acute toxicity tests demonstrated an optimal level of safety; therefore, Cyrene has been defined as “practical non-toxic” to the environment according to the Global Harmonized System of Classification and Labelling of Chemicals (GHS) and readily biodegradable to carbon dioxide and water (99%, 14 days) [4]. In our review, we discuss the properties of the green solvent Cyrene and its applications in organic reactions for the construction of chemical bonds that lead to bioactive molecules, as well as its use for the production of functional materials with biological applications.

## 2. Cyrene: Chemical-Physical Properties, Synthesis and Reactivity

Dihydrolevoglucosenone, commonly known as Cyrene, is a bicyclic acetal [5], derived from cellulose-based biomass. As common acetals, it is stable in basic conditions, while showing marked reactivity towards strong reducing and oxidizing agents and drastic acidic environments [6]. The physical properties of Cyrene in comparison to other common dipolar aprotic solvents are described in Table 1. At room temperature, Cyrene is a colorless viscous liquid (density = 1.25 g/mL) with a high boiling point (227 °C). It has been considered a dipolar aprotic solvent, with polarity parameters (measured by Kamlet–Taft *π**) comparable to NMP (*N*-methyl pyrrolidone), and a significant miscibility with water [7].

The chemical-physical features of Cyrene, combined with high biocompatibility and sustainability issues, enforce its use as an alternative green solvent for organic synthesis (Figure 1). Its high boiling point also makes it a suitable refluxing solvent for carrying out microwave-assisted reactions, and the difficulty in removal from the reaction mixture is easily supplanted by its high miscibility with water, which allows its removal via aqueous work-up and reuse after distillation. Its polarity parameters make it similar to solvents such as NMP and DMF, and its solvent capacity is comparable to that of DMSO; however, unlike DMSO, it does not have the same levels of toxicity and environmental impact and can, therefore, be considered a safer substitute [8,9].

Chemically, Cyrene is a bicyclic ketone that bears an intramolecular acetal functionality, synthetically accessible through two-step processes starting from biomass [10]. Cellulose undergoes a pyrolysis process, affording levoglucosenone which, by catalytic hydrogenation, is converted into dihydrolevoglucosenone (Cyrene) (Figure 1).

The ketone group of Cyrene easily reacts with water to afford the corresponding *gem*-diol, strongly increasing the polarity of the solvent, providing two additional H-donor groups (Figure 2) [11]. The hydration of Cyrene has also been studied in the presence of D_2_O, highlighting that in 1:1 mixtures, it has been observed that hydration is favored, while when an organic co-solvent is present, the ketone form prevails. Therefore, according to the choice of solvent, it is possible to favor the recycling of Cyrene in an aqueous environment (Figure 2a) [11]. Furthermore, the carbonyl group can undergo the formation of cyclic ketals with ethylene glycol, as in the case of the Cygnets derivatives [12] (Figure 2a), forming oximes under the Beckmann reaction [13], and reacts vigorously with organometallic reagents, such as Grignard reagents (Figure 2a). Under drastic basic conditions, Cyrene undergoes intermolecular aldol addition and aldol condensation, leading to derivatives prone to further polymerization (Figure 2b). Cyrene also undergoes Claisen–Schmidt condensation with aldehydes [14], providing spiro-derived exocyclic enones by the double addition of aldehydes to the carbonyl functionality [15,16] (Figure 2b). Finally, it can undergo metal-catalyzed reduction reactions [17] (Figure 2c) and Bayer–Villiger oxidation (Figure 2d) [18].

## 3. Cyrene as a Solvent for Organic Synthesis Transformations

To date, Cyrene has been investigated in numerous organic reactions, as described in Table 2. Below, a description of each type of reaction will be given, basing on the chemical bond formed.

### 3.1. Amide Synthesis

The formation of the amide bond is a relevant synthetic operation because of the frequent occurrence of such a functional group in biologically active molecules. Cyrene was proposed, instead of the classic coupling solvents, such as DMF and NMP, for carrying out amide bond formation reactions starting from acid chlorides [11] and carboxylic acids [19], affording interesting results. Bousfield et al. proposed a simple protocol for the synthesis of amides from acid chlorides in the bio-available solvent Cyrene (Figure 2), reducing wastes and minimizing column chromatography operations [11]. During the optimization, 4-fluorobenzoyl chloride reacted with several pyrrolidines, anilines or benzylamines, using NEt_3_ as the base, affording the corresponding amides in high yield and, in most cases, by simple precipitation via the addition of water, without the need for column chromatography purification. Then, the authors explored the scope of the reaction using different acid chlorides and primary and secondary amines. This procedure avoided the use of toxic solvents, such as DMF and DCM (dichloromethane), and the removal of Cyrene via aqueous work-up resulted in up to a 55-fold increase in molar efficiency (Mol E.%) versus conventional operating protocols.

**Scheme 2 ijms-23-15960-sch002:**
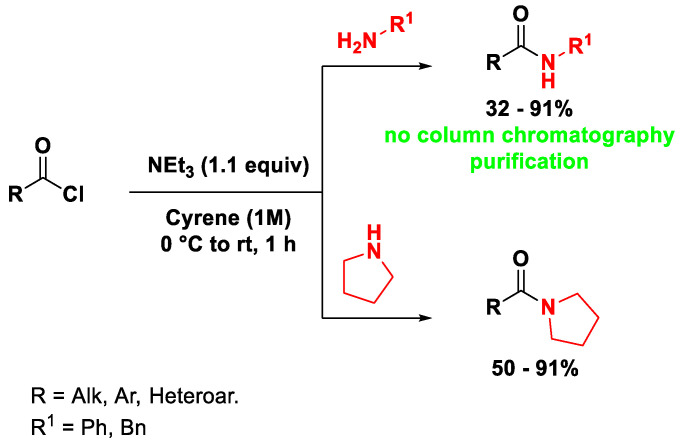
Synthesis of amides from acid chlorides in Cyrene.

The conversion of carboxylic acids to amides using Cyrene as the solvent was investigated by Wilson and co-workers (Figure 3) [19]. They evaluated the employment of Cyrene in the HATU-mediated amide coupling, introducing for the first time such a solvent as a valid alternative to DMF, furnishing 25 examples of lead-like compounds and dipeptides. The best conditions discovered involved the use of HATU/DIPEA coupling agents in Cyrene, under vigorous stirring of the solution due to the high viscosity of the bio-solvent, affording the corresponding products in 1 h at room temperature.

Alternative to DMF, Cabri and collaborators performed solid-phase synthesis (SPPS) using a combination of Cyrene and dimethyl or diethyl carbonate [20]. Although the solubility of the amino acids employed in the study proved to be lower compared to the classic DMF system, the use of the combination of DIC/Oxyma (*N,N′*-diisopropylcarbodiimide/ethyl cyano(hydroxyimino)acetate) coupling reagents was found to be efficient in enhancing the solubility, performing the coupling with optimal results, introducing the use of Cyrene in the SPPS.

### 3.2. Ureas Synthesis

Isocyanates are versatile substrates that can be easily converted to ureas by reacting with amines. Given the great importance of the ureidic bond and its presence in various drugs [21], the search for new biocompatible solvents to be also used on an industrial scale represents an interesting topic. DMF is the most used solvent for this kind of transformation, both for its high solubility properties and for the prolonged reaction time. Camp et al. described the use of Cyrene for the linkage of amines to isocyanate (Figure 4) [22]. They started their investigation by reacting phenylisocyanate with pyrrolidine and precipitating the corresponding urea after treatment with water. Optimization of the reaction conditions led to the formation of a large panel of ureas by reacting isocyanates with several primary and secondary amines, affording the corresponding products in high yield after simple precipitation in water.

### 3.3. Isothiocyanates Synthesis

Isothiocyanates are an interesting scaffold used for the synthesis of thioureas, and their synthesis involves often the use of highly toxic reagents such as thiophosgene or CS_2_. Nickisch and collaborators employed, for the first time, a multicomponent reaction including isocyanides, elemental sulfur and several amines for the successful synthesis of isothiocyanates, in the presence of catalytic amounts of DBU (1,8-diazabicyclo[5.4.0]undec-7-ene) (down to 2 mol%) (Figure 5) [23]. An optimization of the reaction, in terms of sustainability, was made considering ecofriendly solvents such as Cyrene, and purification by column chromatography was further optimized to generate less waste by maintaining the high purity of the product [23].

### 3.4. Carbon–Carbon Bond Couplings

C–C catalyzed cross-couplings are reactions of considerable importance, employed to build complex molecular architectures. The use of palladium-based catalysts often accompanies such type of transformations, and the choice of the solvent/catalyst combination remains a fundamental task for carrying out the process. Nowadays, alternative green solutions to the classical solvents used for Pd-catalyzed C–C cross-couplings represent an occurring field of study. Cyrene has demonstrated to be a valid alternative to DMF, THF and toluene to carry out Sonogashira and Suzuki–Miyaura reactions. Watson and collaborators described, for the first time, a Sonogashira reaction using Cyrene as an alternative solvent to DMF, also expanding the scope of the reaction to a Cacchi-type annulation (Figure 6) [24]. They investigated the reaction between iodobenzene and phenylacetylene, using the catalyst PdCl_2_(PPh_3_)_2_ and CuI as an additive. Interestingly, the reaction worked better at room temperature, exclusively with the use of an organic base such as NEt_3_, reaching completion in 1–5 h, in comparison with inorganic bases. Inorganic carbonates, especially at higher temperatures, led to intermolecular aldol reactions of Cyrene, failing to perform the transformation. The scope of the reaction was exploited, and the use of Cyrene was demonstrated to be suitable for a large panel of functional groups (Figure 6). Additionally, 5-*endo-dig* cyclization of Sonogashira-coupled *o*-amino and *o*-hydroxyaryl derivatives underwent Cacchi-type annulation to afford corresponding bicyclic compounds in good yields.

Today, the Suzuki–Miyaura cross-coupling represents the most widespread Pd-catalyzed C–C bond-forming reaction in the panorama of organic synthesis, and has found countless applications in the medicinal chemical industry. Most of the Suzuki cross-coupling reactions involve the use of dipolar aprotic solvents, with DMF as the solvent of choice. Additionally, Watson et al. again described for the first time a Suzuki cross-coupling protocol including the bio-derived solvent Cyrene as the reaction medium [25]. Pd(dppf)Cl_2_ was used as a catalyst and cesium carbonate was the best choice among the inorganic bases to perform the transformation (Figure 7). The addition of water was fundamental for reducing the rate of formation of the aldol sub-product, that could be otherwise be easily removed via precipitation and filtration in petroleum ether/ethyl acetate mixtures.

Sangon et al. described a classic Heck reaction between iodobenzene and styrene in triethylamine employing Cyrene, demonstrating improved performances during the process, in comparison to conventional dipolar aprotic organic solvents (Figure 8) [26]. The scope of the reaction was discovered to be substrate-specific, because switching styrene with methyl acrylate led to less efficiency in Cyrene compared to other solvents.

Among Pd-free coupling reactions, the Baylis–Hillman reaction represents an interesting transformation leading to the introduction of a new carbon–carbon bond between the α-position of an activated alkene and a carbon electrophile, such as an aldehyde. The reaction takes place in the presence of a nucleophilic catalyst, such as a tertiary amine (generally DABCO furnishes better results) or a phosphine. Solvent media that are usually employed in this conversion are dipolar aprotic solvents, such as 1,4-dioxane and DMF. The same research group introduced Cyrene as a solvent medium for the Baylis–Hillman reaction as a safer and greener alternative, providing excellent results during the process (Figure 9) [26].

### 3.5. Introduction of Fluorinated Functionalities

Fluorinated moieties are, nowadays, becoming increasingly popular thanks both to the interesting properties that the fluorine atom can impart to the molecule (bioisosterism, biocompatibility, improvement of metabolic parameters, *inter alia*) [27], and the reactivity furnished to the obtained structure which can undergo further functionalization [28]. Among the structural modifications in which fluorine can play a significant role, the difluoromethyl group (-CHF_2_) has increasingly established itself in the chemical and pharmaceutical panorama [29]; therefore, novel methods of difluoromethyl functionalization are becoming necessary. In particular, the use of the (difluoromethyl)trimethylsilane (TMSCHF_2_) reagent, in the presence of a base such as KOH or alkoxyde, currently represents one of the most appealing sources of the difluoromethyl group [30,31,32]. Quero and coworkers combined the use of the newly synthesized CuI/BNNSs (copper iodide nanoparticles/boron nitride nanosheets) supports as heterogenous catalysts for the -CH difluoromethylation reaction of heteroarenes and terminal alkynes, and the green reaction medium Cyrene, instead of the common DMF or NMP (Figure 10) [33]. The protocol was carried out at room temperature affording the products in high yields and a shorter reaction time, in comparison to the previously reported strategies.

Introducing fluorine into an aromatic molecule is a feature of particular interest to the pharmaceutical industry. Aromatic fluorination is a SNAr reaction efficiently performed in polar reaction media. Sherwood et al. proposed the use of Cyrene as a greener alternative to DMF and NMP, observing comparable results (Figure 11) [1].

### 3.6. N-Alkylation Reactions

*N*-alkylation reactions, in particular heterocyclic ones, are among the most used by the pharmaceutical industry to carry out the synthesis of biologically active molecules. Sherwood and collaborators proposed a two-step green synthesis of the antidepressant Bupropion, by carrying out a nucleophilic substitution reaction on a secondary alkyl halide with *tert*-butylamine using Cyrene as the *medium* (Figure 12) [34]. *N*-alkylation reactions are generally carried out in polar aprotic solvents, such as DMF and NMP, and this study showed how Cyrene could also be effectively used as a green substitute in this kind of S_N_2 reaction. The synthetic pathway starts from 3-chloropropiophenone, which is converted into the corresponding secondary bromide in the presence of *N*-bromosuccinimide in ethyl acetate as the solvent, and the obtained product is alkylated in Cyrene on *t*-butylamine and salified with 1 M HCl to afford Bupropion as a hydrochloric salt.

The Menschutkin reaction could be considered another intriguing example of *N*-alkylation on heteroatoms. In particular, such transformation has been practiced to obtain imidazolium-based ionic liquids via alkylation on the imidazole nitrogen with 1-bromodecane. The salt could be easily obtained with Cyrene as the medium with good results, instead of the common NMP, as illustrated by Sherwood and coworkers (Figure 13) [1].

### 3.7. Formylation of Amines

A curious dual role of Cyrene as a solvent–catalyst can be found in the work of Yu and collaborators, who proposed the synthesis of formamides starting from amines and CO_2_ in the presence of phenylsilane as a reductive agent (Figure 14) [35]. Usually, organosilanes are used as reducing agents in combination with catalytic amounts of trifluoroacetic acid (TFA), Pd-, Rh- or boron-based catalysts, i.e., tris(pentafluorophenyl)borane). In this interesting example, the reduction in Cyrene was reported for the first time, without the use of an auxiliary catalyst, affording various substituted formamides. The scope of the reaction was also expanded to the synthesis of benzothiazoles from the reductive conversion of CO_2_ with aminothiophenols with the same reducing agent. The Cyrene-based catalytic system possessed a higher reaction rate constant and a lower apparent activation energy than both γ-valerolactone and acetonitrile-based catalytic systems. Furthermore, the authors demonstrated that Cyrene possessed a strong ability to activate the N–H bond in amines and the Si–H bond in phenylsilane by the solvent effect (i.e., solvation and polarization), thereby enhancing the catalytic efficiency.

### 3.8. Microwave-Assisted Multicomponent Reactions

The use of Cyrene has also been found in microwave-assisted synthesis, in particular as a substitute for DMF, ensuring better performance by being able to heat to higher temperatures. Interesting work published by Tamargo and collaborators [36] clearly demonstrated the relevance of using a multicomponent microwave process in Cyrene as a solvent for the synthesis of bipyridine analogues (Figure 15). Starting from chromone derivatives, aliphatic or aromatic amines and phenyl or pyridyl acetonitrile, it was possible to assemble several bipyridine analogues in a multicomponent reaction; in particular, the use of Cyrene and higher temperatures (150 °C) in a microwave-assisted protocol resulted in impressive yields. For simplicity, we report an example of the transformation using aniline, 3-formyl-chromone and pyridinyl acetonitrile (Figure 15). Bipyridine analogues represent one of the most relevant chemical classes that can be found in many natural products and functional materials. The authors also highlighted the function of their products to bind Hg^2+^, Fe^3+^ and Cu^2+^, which enables bipyridine analogues employed for several applications, such as heavy metal chelating agents and pollution monitoring.

### 3.9. Biocatalyzed Reduction of α-Ketoesters to α-Hydroxyesters

Biocatalytic transformations involve the use of enzymes to perform chemical transformations on organic compounds. Water remains the most relevant medium for biocatalyzed transformations, but due to its properties, the presence of organic (co)solvents is sometimes necessary. Therefore, the reduction of α-ketoesters to α-hydroxyesters was hardly performed in water, because of the concomitant hydrolysis of the reactant. De Gonzalo optimized the reduction of α-hydroxyesters in Cyrene, employing the biobased solvent for the first time in a biocatalyzed processes, using alcohol dehydrogenase as a catalyst (Figure 16) [37]. A set of α-hydroxyesters was obtained from the corresponding chiral α-keto derivatives, with high yields and optical purities. In addition, nicotinamide co-factor or glucose dehydrogenase were used as recycling systems. It was worth noting that Cyrene’s ketone group remained unaltered during the process at the concentration used (up to 30% *v*/*v*).

The use of Cyrene was also investigated in esterification and hydrolysis reactions, as documented by the work of the Guajardo group in 2020 (Figure 17) [38]. A monoesterification of glycerol with benzoic acid was carried out using the Novozym 435 lipase or crosslinked aggregates of lipase B from *Candida antarctica* (CAL-B), in the presence of a small amount of water. Cyrene has been shown to be an optimal solvent for the transformation, both for its high solvent capacity (comparable to 1,4-dioxane) and for the recycling ability of the CAL-B catalyst. The instability of Cyrene in acid can be circumvented by increasing the concentration of glycerol. The opposite reaction was carried out using the same enzyme, employing concentrations of Cyrene below 40%: by increasing the concentration of Cyrene with respect to water, the performance of the hydrolysis decreased due to the *equilibrium* shifting towards the starting material.

Unfortunately, the esterification of stearic acid with beta-sitosterol was unsuccessful [39], and Cyrene also failed to carry out the esterification of 2-phenylpropionic and cinnamic acids with ethanol under the same lipase-catalyzed reaction [40].

## 4. Cyrene as an Innovative Solvent for the Preparation of Nanomaterials with Advanced Applications

### 4.1. Nanoparticles as Drug Delivery Systems

The use of nanotechnology and, more specifically, nanoparticles (NPs) as drug delivery systems has gained much attention in the past few decades and constitutes a powerful strategy to overcome the poor solubility, stability and biocompatibility of many bioactive molecules [41].

In recent years, biodegradable polymeric nanoparticles have been widely used as potential drug delivery devices because of their high biocompatibility, long circulation time and controlled drug release. Many techniques have been developed for the preparation of those polymeric nanoparticles, but they suffer from the use of toxic and environmentally harmful organic solvents, such as ethyl acetate, chloroform, and dichloromethane.

The use of alternative no-toxic solvents has been reported, but they present many disadvantages, such as poor water solubility, high viscosity and plasticizing effects [42]. To overcome these limits, Grune et al. proposed the use of Cyrene as a new sustainable and innovative solvent for the preparation of poly(lactic-co-glycolic acid) nanoparticles (PLGA-NPs) [43].

PLGA-NPs with a small hydrodynamic diameter (220 nm), a low polydispersity index (<0.2) and a highly negative zeta potential (<−23 mv) were, therefore, obtained by dissolving Resomer RG502, chosen as a standard polymer, in Cyrene. The organic solution was then covered with 2% aqueous poly(vinyl alcohol) (PVA), followed by ultrasonication (30 s at a cycle of 100%) and magnetic stirring of the NP suspension for 2 h at room temperature.

As mentioned before, the aprotic Cyrene is highly soluble in water because of the presence of an equilibrium between keto and hydrate forms. This leads to the rapid diffusion of Cyrene to the water phase, the precipitation of the polymer and the formation of PLGA-NPs (Figure 3).

After centrifugation, highly purified NPs were obtained with a residual Cyrene content of <2.5%.

Furthermore, in order to test the use of such NPs as drug delivery systems, the lipophilic drug atorvastatin was encapsulated by simple dissolution in Cyrene. The encapsulation was achieved in high yields and without modifying the physicochemical properties of the NPs. Moreover, an in vivo study carried out with the hen’s egg model demonstrated the high biocompatibility of Cyrene and its NPs.

Later on, the same group applied the same strategy to encapsulate 6-bromoindirubin-3′-glycerol-oxime ether (6BIGOE) into PLGA-based NPs, which is a potent anti-inflammatory derivative of the natural product indirubin suffering from poor water solubility, instability and cytotoxicity [44].

Once again, they demonstrated that Cyrene was able to assist a fast and convenient method for the preparation of drug-loaded NPs with no toxic and plasticizing effects in comparison with standard methods.

### 4.2. Graphene-Based Nanomaterials

Nanoscale two-dimensional graphene materials have attracted remarkable attention in many fields because of their unique proprieties, such as biocompatibility, infrared light absorbance, thermal conductivity, high surface area and easy modification.

The use of these novel nanomaterials for biomedical applications has grown exponentially in recent years, and they have been employed as bacterial inhibitors, imaging contrast agents and drug/gene stimuli-sensitive delivery systems able to respond to specific external or internal stimuli (i.e., temperature, light, pH, redox and enzymes) [45]. Moreover, many anticancer drugs, proteins and genes have demonstrated a higher efficiency when combined with graphene-based materials. Liquid exfoliation (and dispersion) from graphite is the preferred scalable method to prepare good quality graphene; however, low yields are obtained with this procedure and the use of toxic and non-environmentally friendly solvents such as NMP and DMF is required. Therefore, in order to solve performance and safety issues, a new approach was developed by Salavagione et al. by using the new renewable solvent Cyrene [46]. They focused on a fast processing time where solvent (3 mL) was added to a vial containing graphite (4.5 mg); the mixture was treated with an ultrasonic probe for 15 min and the dispersion was then centrifuged for 10 min (7000× *g* rpm). High concentrations of graphene dispersions were obtained in comparison with traditional solvents (Figure 4). These results may be referred to the near-optimal Hansen solubility of Cyrene, as well as its high viscosity. In addition, graphene produced in Cyrene was of significantly higher quality, with 93% of flakes comprising 10 layers or less, as well as having a greater aspect ratio and much lower edge defects.

Later on, Pan et al. also studied the use of Cyrene for the dispersion of graphene using the same sonication method [47]. In particular, they compared the conductivity properties of the graphite flakes obtained using Cyrene as opposed to NMP. Cyrene was found to be more effective since it required less time to achieve the same results of conductivity. Moreover, the graphite flakes were characterized and evaluated by atomic force microscopy (AFM) and the stable existence of few-layer graphene nanoflakes was confirmed.

### 4.3. Nano–Sized Metal Organic Frameworks (MOFs)

Metal organic frameworks (MOFs) are a family of hybrid porous materials made of well-organized inorganic metal cations, clusters or neutral metals (nodes) linked to one another by organic ligands (linkers) via coordination bonds (Figure 5). A large number of MOFs has been synthesized in the past decades, and the driving force of the enormous interest in this area lies in the versatility of the metal-organic coordination chemistry, as well as the wide range of available organic linkers which can lead to thousands of possible structures (typically within the 3–60 Å range). With this in mind, researchers have tuned MOFs toward a wide range of applications, including gas adsorption and separation, catalysis, energy storage and biomedical applications. Within this framework, MOFs have recently been reported as efficient materials for drug delivery systems using Ibuprofen as a model drug [48]. In fact, these solids exhibit many interesting features suitable for their successful use as drug carriers, such as a low toxicity (the use of low toxicity metals and ligands), biodegradability, the presence of a hydrophilic–hydrophobic internal microenvironment adapted to a large variety of active molecules, modulation of drug encapsulation and delivery by controlling the porous structure (interconnectivity, pore size, flexibility), and optimization of host–guest interactions.

MOFs are often synthesized at elevated temperature using toxic organic solvents, such as DMF or diethylformamide (DEF), due to their high boiling point, as well as their beneficial acid–base properties. However, the industrial-scale synthesis of MOFs may generate significant amounts of DMF waste, which exhibit reprotoxicity and end-of-life issues associated with the formation of NOx upon incineration. Therefore, it is crucial to develop green and sustainable methods of synthesizing MOFs. Thus, in 2016, Katz and co-workers investigated the use of Cyrene in the synthesis of archetypal MOFs [49]. Firstly, they explored the use of a mixture of Cyrene and EtOH for the synthesis of MOF HKUST-1 which was successfully obtained as a crystalline solid. They then proceeded to evaluate its N_2_ gas absorption., observing a decrease in the BET surface area (SA) when Cyrene was used instead of DMF under identical conditions (Table 3). Hence, they investigated the role of EtOH as an additive and found that the MOF’s SA decreased when absolute ethanol was not used. This may be attributed to the fact that in the presence of water, Cyrene might react to produce the geminal diol, as mentioned before. Thus, the use of dry ethanol was obligatory, as well as the necessity of keeping the water content of Cyrene low [49].

With these results in mind, the use of Cyrene was extended to other MOFs: UiO-66, ZIF-8, MOF-74 and Zn(BCD)_2_ (DABCO), as representatives. Both DMF and Cyrene gave the same products in all four cases; however, Cyrene-MOFs show lower surface areas in comparison to their DMF counterparts (Table 4). In order to explain this, the authors looked more closely at ZIF-8: powder X-ray diffraction showed different peak intensities which were not usually observed for pore-filled ZIF-8. Fortunately, examination under a microscope revealed the presence of an aldol condensation product between two Cyrene molecules which could be avoided by decreasing the heating time (Figure 18). All these considerations can be extended to the synthesis of other MOFs which, at present, do not have green synthetic pathways.

### 4.4. Lignin-Based Nanomaterials

Lignin is the second most abundant component of biomass which features unique properties, such as antioxidant, antibacterial, anti-UV properties, and good biocompatibility. Therefore, it represents an optimal candidate for various biomedical applications, such as gene and drug delivery [50]. Several types of lignin exist which are classified according to their production methods; among them are kraft (KL), organosolv (OSL) and soda lignin (SL). Very few organic solvents are indeed able to dissolve a large range of technical lignins; polar aprotic solvents such as DMF and DMAc are among them. However, as before mentioned, they present serious health and environmental issues; therefore, there is a growing need to look for safer and greener strategies. Thus, in 2020, Ragauskas and co-workers examined, for the first time, the use of Cyrene as an alternative polar aprotic solvent to extract lignin [51]. The results showed that Cyrene has a high potential as a solvent in terms of lignin fractionation/recovery. Moreover, their study also showed a good solubility of organosolv lignins in a Cyrene–water mixture (from 60 to 100 vol% Cyrene). Based on these results, Averous et al. expanded the potential of Cyrene to other type of lignins [52]. In agreement with the previous results, OSL was found to be almost fully soluble in Cyrene, while KL and SL were partially soluble (61–63%). The solubility increased when water was added to Cyrene due to the formation of the geminal diol (*vide supra*). Then, they successfully used Cyrene as a solvent for the chemical modification of lignins though the reaction with cyclic anhydride, acyl chloride and isocyanate (Figure 19). To the best of our knowledge, this is the first example of urethane formation in Cyrene; however, further optimization would be needed. This study proves that Cyrene is a valid alternative for greener lignin chemistry.

## 5. Conclusions

To conclude, Cyrene is acquiring more popularity among green, non-mutagenic and non-toxic reaction solvents, thanks to its biocompatibility and applicability in the panorama of organic synthesis and the development of functional biomaterials. Various reactions have been exploited in Cyrene, most of them with high importance for the construction of biologically relevant chemical bonds. On the other hand, Cyrene has also proven its worth for the synthesis of nanomaterials having many advanced applications. Therefore, new interesting frontiers for the use of Cyrene in other fields are opening currently and its use will hopefully become more widespread.

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
