# Peer review of "Cyrene: A Green Solvent for the Synthesis of Bioactive Molecules and Functional Biomaterials"

_ijms, 2022, doi:10.3390/ijms232415960_

Round 1

Reviewer 1 Report

 The manuscript presents a review on the green solvent Cyrene – its preparation, physico-chemical properties, reactivity and organic reactions carried out in Cyrene as a medium. Furthermore, an overview of the Cyrene-assisted preparation of advanced nanomaterials, such as graphen-, MOFs-, lignin-based structures, is described. The collected literature resources are well analyzed, classified and presented in clear and easy-understandable manner.

In my opinion, the authors should improve their manuscript before publication. Here are my comments that may help them:

1. A great advantage of Cyrene is its low toxicity. In the text, it is emphasized many times, for example in lines 73-74, where a suitable reference(s) on the evidence for non-toxic and non-mutagenic properties is needed.   

2. lines 89-90: “Furthermore, carbonyl group can undergo the formation of cyclic ketals with ethylene glycol, as in the case of the Cygnets derivatives [9] (Figure 2),…”. This reaction is not given in Fig 2, so please correct accordingly.

3. lines 96-97: “It can undergo metal reduction reactions.” Most probably the authors mean metal-catalyzed reduction of Cyrene, but the phrase is not correct in the context of given reaction.

4. line 102: Please, replace Table 1 with Table 2.

5. line 112: please add a reference after “(Scheme 2)”

6. In Section 3.3., it is reported a paper by Nickisch et al. (RSC Adv., 2021,11, 3134), which is not mentioned in the reference list.

7. In Scheme 5, please, draw the functional groups structurally, like in other schemes.

8. line 235-236 – unclear sentence.

9. line 244 – replace Buproprion with Bupropion

10. lines 426-427; Ref 42 is wrongly cited here. Seems that the paper by Kats et al. (2016) is not given. Please, check it.

11. line 438: please add an appropriate reference. Overall, please check again all references.

12. There are many abbreviations throughout the text, some of them are not introduced at all (for example DCM (line 118), DIC/Oxyma (line 135), DBU (line 157), TMSCHF2 (line 223), BNNS (line 226) but given as BNNSs (in scheme 10), NSB (line 250), TFA (line 266), while other once denoted should be used further (for example, in lines 457-458 DMF and DMAc). Also, check DEF in line 422.

Author Response

Reviewer#1

Comments and Suggestions for Authors

The manuscript presents a review on the green solvent Cyrene – its preparation, physico-chemical properties, reactivity and organic reactions carried out in Cyrene as a medium. Furthermore, an overview of the Cyrene-assisted preparation of advanced nanomaterials, such as graphen-, MOFs-, lignin-based structures, is described. The collected literature resources are well analyzed, classified and presented in clear and easy-understandable manner.

In my opinion, the authors should improve their manuscript before publication. Here are my comments that may help them:

- We appreciated the very positive feedback of the Reviewer and we are thankful for his/her suggestions to further improve our article. Here they are point by point responses to the comments.

  1. A great advantage of Cyrene is its low toxicity. In the text, it is emphasized many times, for example in lines 73-74, where a suitable reference(s) on the evidence for non-toxic and non-mutagenic properties is needed.  

- References pertaining the remark of the low toxicity of cyrene have been added as indicated (see Ref.#8 and Ref.#9). 

  1. lines 89-90: “Furthermore, carbonyl group can undergo the formation of cyclic ketals with ethylene glycol, as in the case of the Cygnets derivatives [9] (Figure 2),…”. This reaction is not given in Fig 2, so please correct accordingly.

- Corrected accordingly (see box 2a).

  1. lines 96-97: “It can undergo metal reduction reactions.” Most probably the authors mean metal-catalyzed reduction of Cyrene, but the phrase is not correct in the context of given reaction.

- Yes, that’s what we meant. The phrase have been corrected accordingly.

  1. line 102: Please, replace Table 1 with Table 2.

- Done.

  1. line 112: please add a reference after “(Scheme 2)”.

- Reference has been added as requested (see Ref.#11).

  1. In Section 3.3., it is reported a paper by Nickisch et al. (RSC Adv., 2021,11, 3134), which is not mentioned in the reference list.

- Yes, you’re right. It was mistakenly omitted in the final version of the manuscript. Now it is inserted in the references list (see Ref.#23).

  1. In Scheme 5, please, draw the functional groups structurally, like in other schemes.

- Scheme 5 revised accordingly. Functional groups for this type synthesis are really too many (please, check the related reference) and we think it is not the case to draw all. With “R = Alk, Ar, Bn” we tried to stay general as our best.

  1. line 235-236 – unclear sentence.

- The sentence has been re-written for clarity.

  1. line 244 – replace Buproprion with Bupropion.

- Corrected accordingly.

  1. lines 426-427; Ref 42 is wrongly cited here. Seems that the paper by Kats et al. (2016) is not given. Please, check it.

- Checked and corrected.

  1. line 438: please add an appropriate reference. Overall, please check again all references.

- An appropriate reference has been added at this point as requested and the overall list of the references revised.

  1. There are many abbreviations throughout the text, some of them are not introduced at all (for example DCM (line 118), DIC/Oxyma (line 135), DBU (line 157), TMSCHF2 (line 223), BNNS (line 226) but given as BNNSs (in scheme 10), NSB (line 250), TFA (line 266), while other once denoted should be used further (for example, in lines 457-458 DMF and DMAc). Also, check DEF in line 422.

- All abbreviations have been introduced and reported in the main text as indicated.

Reviewer 2 Report

The manuscript entitled “Cyrene: a green solvent for the synthesis of bioactive molecules and functional biomaterials” describes the opportunities of using Cyrene in organic synthesis. Organic reactions carried out in Cyrene medium are reviewed. I found it suitable for publication after minor revisions.

1. Prepare a table that describes the advantages and disadvantages of Cyrene.

2. Scheme 4, the reagents of the reaction should be presented.

3. Scheme 7, some examples of aryl halides and boronic acids should be provided.

4. Add a section about the toxicity of Cyrene and review some available data about “half maximal effective” and “half maximal inhibitory” concentrations.

Author Response

Reviewer#2

Comments and Suggestions for Authors

The manuscript entitled “Cyrene: a green solvent for the synthesis of bioactive molecules and functional biomaterials” describes the opportunities of using Cyrene in organic synthesis. Organic reactions carried out in Cyrene medium are reviewed. I found it suitable for publication after minor revisions.

- We thank the Reviewer for the appreciation of our article. Here they are point by point responses to each minor issue raised.

  1. Prepare a table that describes the advantages and disadvantages of Cyrene.

- We tried to sum up the most important advantages of Cyrene in Figure 1. Now we completed the Figure 1 by adding also the disadvantages as requested.

  1. Scheme 4, the reagents of the reaction should be presented.

- The reaction depicted in the Scheme 4 involves only various isocyanates and the reported amine for the formation of the related ureas.

  1. Scheme 7, some examples of aryl halides and boronic acids should be provided.

- Requested examples have been provided as suggested.

  1. Add a section about the toxicity of Cyrene and review some available data about “half maximal effective” and “half maximal inhibitory” concentrations.

- We took into consideration this observation and we added a brief description about the toxicity of Cyrene within the Introduction section. Based on our knowledge, no data about “half maximal effective” and “half maximal inhibitory” are available for Cyrene. So, we think there is no enough literature for a full section on this topic.

Reviewer 3 Report

The manuscript introduced the most representative applications of cyrene as a green solvent in organic synthesis. The types of the reaction are extensive and the references are appropriate and latest, I think the scientific impact, as well as the overall quality of the manuscript can meet the requirement of the IJMS, and it is suitable for publication after some minor issues are addressed.

Suggestions:

1.The Figure 2 could be divided into 4 parts (such as Figure 2a, Figure 2b, 2c and 2d.) and to discuss respectively in the text, which is more convenient for the reader (from line 85-98).

2.Line 102, “Table 1” should be “Table 2”;

3.The presentation for “Functional group” of “Suzuki-Miyaura” reaction in Table 2 (R1-R2) is inappropriate since the “R” is not a functional group.

4.Line 160, giving the reference number should be enough.

5.Line 250, the “NSB” should be “NBS”;

Author Response

Reviewer#3

Comments and Suggestions for Authors

The manuscript introduced the most representative applications of cyrene as a green solvent in organic synthesis. The types of the reaction are extensive and the references are appropriate and latest, I think the scientific impact, as well as the overall quality of the manuscript can meet the requirement of the IJMS, and it is suitable for publication after some minor issues are addressed.

  • We thank the Reviewer #3 for the very positive feedback which makes us eager to meet his additional requests in order to further improve the quality of the article. Here they are point by point responses to each issue raised.

Suggestions:

  1. The Figure 2 could be divided into 4 parts (such as Figure 2a, Figure 2b, 2c and 2d.) and to discuss respectively in the text, which is more convenient for the reader (from line 85-98).

- Figure 2 has been revised as requested and the main text corrected accordingly.

  1.  Line 102, “Table 1” should be “Table 2”.

            - Corrected.

  1. The presentation for “Functional group” of “Suzuki-Miyaura” reaction in Table 2 (R1- R2) is inappropriate since the “R” is not a functional group.

            - Table 2 has been revised with proper functional groups.

  1. Line 160, giving the reference number should be enough.

           - Corrected accordingly.

  1.  Line 250, the “NSB” should be “NBS”;

          - Corrected. We actually put directly N-bromosuccinimide as it is repeated

           in the main text.